# Antimicrobial Consumption in the Livestock Sector in Bhutan: Volumes, Values, Rates, and Trends for the Period 2017–2021

**DOI:** 10.3390/antibiotics12020411

**Published:** 2023-02-18

**Authors:** Ratna B. Gurung, Karma P. Zangmo, James R. Gilkerson, Glenn F. Browning, Angeline S. Ferdinand, Mauricio J. C. Coppo

**Affiliations:** 1National Centre for Animal Health, Serbithang, Department of Livestock, Ministry of Agriculture and Forests, Thimphu 11001, Bhutan; 2Asia-Pacific Centre for Animal Health, Melbourne Veterinary School, Faculty of Science, University of Melbourne, Parkville, VIC 3010, Australia; 3Centre for Health Policy, Melbourne School of Population and Global Health, University of Melbourne, 207 Bouverie St., Carlton, VIC 3053, Australia; 4Escuela de Medicina Veterinaria, Facultad de Ciencias de la Vida, Universidad Andrés Bello, Concepción 4300866, Chile

**Keywords:** antimicrobial consumption, livestock, population correction unit, South Asia, antimicrobial stewardship, pcu

## Abstract

Data on the use of antimicrobials in humans and livestock may provide evidence to guide policy changes to mitigate the risk of antimicrobial resistance (AMR). However, there is limited information available about antimicrobial use in livestock in low- and middle-income countries, even though these nations are most vulnerable to the impact of AMR. This study aimed to assess the consumption of veterinary antimicrobials in Bhutan and identify areas for improvement to reduce the use of antimicrobials in livestock. National data on livestock numbers and annual procurement of veterinary antimicrobials over five years (2017–2021) were used to calculate rates of antimicrobial consumption and annual national expenditure on veterinary antimicrobials in Bhutan. The rate of antimicrobial consumption in Bhutan was 3.83 mg per population correction unit, which is lower than most countries in Europe, comparable with the rates of consumption in Iceland and Norway, and approximately 120-fold lower than published rates of antimicrobial consumption in South Asian countries, including Nepal and Pakistan. The low rates of antimicrobial consumption by the animal health sector in Bhutan could be attributable to stronger governance of antimicrobial use in Bhutan, higher levels of compliance with regulation, and better adherence to standard guidelines for antimicrobial treatment of livestock.

## 1. Introduction

In Bhutan, the agriculture sector is the main source of employment, with 51.1% of the population engaged in crop and livestock farming [1]. Livestock farming contributes 4% to the national gross domestic product (GDP), and an estimated 22% of rural household income is generated from rearing livestock [2]. The predominant livestock species are cattle and poultry [3,4,5,6,7]. The animal health service, including supply of veterinary medicines, is considered a state good provided free of cost to all livestock farming communities [8]. The National Centre for Animal Health (NCAH), under the Department of Livestock, procures veterinary medicines, including antimicrobials, based on annual estimates of requirements. The NCAH also collects and records data annually on the value and volume of veterinary medicines procured and distributed for the whole country. Bhutan imports all its human and veterinary medicines. Use of veterinary medicines, including antimicrobials, is regulated by the Drug Regulatory Authority (DRA) of Bhutan, and the use of veterinary and human medicines is governed by the Medicine Act of the Kingdom of Bhutan 2003 and the Bhutan Medicines Rules and Regulations 2019. To be eligible to supply any medicine, suppliers have to register the product with the DRA of Bhutan. Upon registration, the DRA oversees the approval process that leads to authorization of importation [9]. This enables the DRA to ensure the availability of quality and safe medicinal products in the country [10].

The acquisition of antimicrobial resistance (AMR) enables microorganisms to persist or grow in the presence of drugs designed to inhibit or kill them. The underlying driver of the development of AMR is inappropriate use of antimicrobial drugs, which is contributed to by inadequate dosing, use of the wrong antimicrobial, use for the incorrect duration, and weak regulation of use. Such practices facilitate and expedite selection for resistance. Antimicrobial resistance is a well-recognized problem and a global threat of increasing concern because of its implications for food safety, food security and the economic well-being of millions of farming communities, as well as human and animal health and welfare. Because of the serious implications of AMR for livestock production systems, many countries have started looking at trends in antimicrobial consumption, good prescribing practices, and the effectiveness of existing regulations and have begun developing strategies to mitigate the development of AMR. In an effort to monitor the consumption of antimicrobials by livestock, many countries have initiated recording of their use and publication of these data annually. The annual antimicrobial consumption rate by livestock in New Zealand was 10.40 mg per population correction unit (mg/PCU) in 2018 [11]. Similarly, the livestock sectors in 31 European nations consumed an average of 89 mg of antimicrobials/PCU, with the lowest consumption seen in Norway (2.30 mg/PCU) and the highest in Cyprus (393.9 mg/PCU) [12]. In 2010, the world’s five largest antimicrobial consumers in livestock were China, the USA, Brazil, Germany, and India, and the five largest consumers in 2030 are projected to be China, the USA, Brazil, India, and Mexico [13]. Although data on consumption of antimicrobials by livestock are available from many developed nations, there is still limited information from low- and middle-income countries in Africa and Asia, even though these are the nations that are most vulnerable to the impact of AMR because of their low gross domestic product (GDP) per capita [14]. Therefore, it is imperative that these nations also monitor and evaluate the use of antimicrobials in humans and livestock. The findings from such evaluations may provide evidence to guide policy changes and decision making to mitigate the risk of AMR.

This study aimed to assess the consumption of veterinary antimicrobials in Bhutan in terms of volume, value, and rate over last five years (2017–2021), to examine the trends in consumption, and to identify areas for improvement to reduce the use of antimicrobials in livestock production systems.

## 2. Materials and Methods

### 2.1. Livestock Population

Livestock population data for the years 2017–2021 were accessed from the National Statistics Bureau of Bhutan [3,4,5,6,7]. The livestock populations included in the study were bovines (cattle, yaks, mithun, and buffalos) and sheep, goats, pigs, and poultry (chickens and turkeys). The total number of individuals of each animal species was calculated for each of the five years.

### 2.2. Population Correction Unit

The population data were used to calculate the number of population correction units (PCU) of each type of livestock. PCUs are a measure of livestock populations that accounts for the differences in body weight between different animal species. The numbers of PCUs were derived using a method developed by the European Medicines Agency [12] and the tool used by the European Surveillance for Veterinary Antimicrobial Consumption (ESVAC) project. They were calculated by multiplying the standard average weight of the species and type of animal by the country’s population of that animal species and type over each year. The standardised average weights of each animal species and type, as provided by ESVAC, are shown in Table 1.

### 2.3. Antimicrobial Procurement and Consumption

National data on annual procurement of veterinary antimicrobials were accessed from the National Centre for Animal Health of the Department of Livestock. All antimicrobials were grouped into their respective classes. The amount of active constituent in each antimicrobial preparation was determined and used to calculate the total mass of active antimicrobial. Antimicrobial consumption was calculated as milligrams (mg) of antimicrobials per PCU of animals (mg/PCU). The consumption of veterinary antimicrobials in Bhutan for 2020 was then compared with that of 31 European nations, other regional countries, and New Zealand [11,12].

### 2.4. Cost of Antimicrobials Consumed

The cost of each antimicrobial was collected and compiled. The cost in Bhutanese currency—the Bhutanese ngultrum (BTN)—was converted into US dollars (USD) at the exchange rate of USD 1 to BTN 79.5.

## 3. Results

### 3.1. Livestock Populations

Over the last five years, there was a steady increase in the livestock population in Bhutan (Table 2). This increase is mainly attributable to an increase in the number of poultry enterprises in the country. The most common livestock species in Bhutan are ruminants and poultry. The pig, goat, sheep, and horse populations did not increase over the period considered in this study.

### 3.2. Population Correction Units

The standardised average weight of the different livestock species and types (Table 1) and the livestock populations (Table 2) were used to calculate the total biomass of animals and the total number of PCU for each livestock species (Table 3).

### 3.3. Antimicrobial Consumption

#### 3.3.1. Classes of Antimicrobials

Sulphonamides were the class of antimicrobials that was most heavily used in Bhutanese livestock, followed by tetracyclines, and then other classes of antimicrobials (metronidazole and nitrofurazone) (Table 4). Considerable amounts of tetracyclines were procured and administered in 2017 and 2018. In the absence of accurate information about the ultimate use of tetracyclines, we presume that they were used in poultry, as this class of antimicrobial was available as a 5% powder in 100 g sachets, a convenient size and formulation for administration in feed or water to treat sick birds. However, the trend over the 5 years examined in this study was a decline in use of sulphonamides and tetracyclines. The proportions of each class of antimicrobials within the annual total amounts consumed are shown in Figure 1. Sulphonamides are often used concurrently with trimethoprim as potentiated sulphonamides. Consumption of aminoglycosides, amphenicols, fluoroquinolones, and penicillins did increase significantly over the five-year period. Use of first- and second-generation cephalosporins reduced dramatically, while the use of third- and fourth-generation cephalosporins increased by more than threefold. It appears that use of first- and second-generation cephalosporins has been replaced by use of third- and fourth-generation cephalosporins. Macrolides (erythromycin) were only used in 2017 and only in very small quantities in combination with tetracyclines and a vitamin mixture powder. Similarly, colistin was only used in 2017 as a powder also containing amoxicillin trihydrate. Enforcement of restrictions on the use of these antibiotics in livestock resulted in discontinuation of their use in 2018 and thereafter.

#### 3.3.2. Annual Antimicrobial Consumption in the Livestock Sector

The calculations of the rate of consumption of antimicrobials in this study include horses along with other livestock species, even though horses are not consumed as meat in Bhutan (Table 5). The average annual rate of consumption of antimicrobials by livestock, including horses, for the period 2017–2021 was found to be 4.09 mg/PCU (with a range from 3.81–4.55). Horses were included to enable a direct comparison with published data from European nations, which included horses. If horses are excluded, the rate of consumption was 4.32 mg/PCU (4.03–4.80). There was no appreciable change in consumption over the 5-year period, whether horses were included or not, possibly because of the relatively small size of the horse population.

Annually, sulphonamides and potentiated sulphonamides were used at the highest rate (1.43 mg/PCU), followed by tetracyclines (1.29 mg/PCU) (Table 6). Consumption of the other classes of antimicrobials was <1 mg/PCU.

#### 3.3.3. Comparisons of Rates of Antimicrobial Consumption in Livestock in Bhutan with Those in Other Countries

The data on the rates of antimicrobial consumption in livestock in Bhutan in 2020 were compared with those of 31 European nations (2020 data), other regional countries, and New Zealand. The rate of consumption in Bhutan was 3.83 mg/PCU, which was comparable with the rates of consumption in Iceland and Norway (Appendix A). The size of the animal population of Iceland (in PCU) was similar to that of Bhutan, as was its rate of antimicrobial consumption and total volume of antimicrobials consumed. In the South Asian region, Nepal and Pakistan have reported consumption rates in chickens of 500 mg/PCU [15] and 462.57 mg/PCU) [16], respectively.

In many of the countries included in this comparison, penicillins were the class of antimicrobials most heavily used in livestock, followed by tetracyclines and sulphonamides/trimethoprim (Appendix A). Many countries have moved away from or severely restricted the use of first- and second-generation cephalosporins, but have started using greater amounts of higher-generation cephalosporins. Similarly, in 2020, there was negligible use of first- and second-generation cephalosporins in livestock in Bhutan, but there was a notable increase in the use of third- and fourth-generation cephalosporins. Overall, the rates of consumption by livestock of antimicrobials by class in Bhutan were comparable to those of Iceland and Norway.

### 3.4. Annual Expenditure on Veterinary Antimicrobials

The costs in USD of the annual consumption of veterinary antimicrobials in Bhutan over the last five years are shown in Table 7. Over the last five years, the annual expenditure on antimicrobials for livestock ranged from USD 59,135 to 73,927. The highest expenditure over the five-year period from 2017 to 2021 was on the potentiated sulphonamides, followed by penicillins, tetracyclines, cephalosporins, and aminoglycosides.

## 4. Discussion

The rates of antimicrobial consumption by livestock in Bhutan were found to be comparable with those of the lowest-consuming countries in Europe. The rate of consumption by the livestock sector (3.83 mg/PCU in 2020) was well below the average rate of consumption of European nations (89 mg/PCU) [12] and less than half that of New Zealand (10.4 mg/PCU) [11]. In the South Asian region, the rate of consumption in Bhutan was far lower than that of Nepal (500 mg/PCU in chickens) [15] and Pakistan (462.57 mg/PCU in chickens) [16]. In Nepal, 13% of poultry producers use antimicrobials as growth promoters [17], but the Animal Feed Standard of Bhutan prohibits the use of antimicrobials in feed as growth promoters or for prophylaxis or metaphylaxis. The low rate of consumption of antimicrobials by the animal health sector in Bhutan could be attributable to the following:The governance of antimicrobial use in Bhutan (in both human and animal health) by the Medicine Act of the Kingdom of Bhutan 2003 and the Medicine Rules and Regulations of Bhutan 2019. The legal provisions in this act and these rules and regulations are regulated by the Drug Regulatory Authority of Bhutan. There is an appreciable level of compliance with these regulations by all stakeholders;Guidelines in the livestock sector to ensure appropriate use of antimicrobials in livestock production systems. These include Standard Treatment Guidelines, Antibiotic Treatment Guidelines, draft Infection Prevention and Control Guidelines, the Animal Feed Standard, and the National Veterinary Drug Formulary;Veterinary antimicrobials in Bhutan are prescription-only medicines. Only certified veterinarians can prescribe antimicrobials. No antimicrobials are available over-the-counter.

Antimicrobial stewardship programs have not yet been incorporated into veterinary care in Bhutan. Initiation of an antimicrobial stewardship program might be expected to focus efforts to reduce, replace, and refine the use of antimicrobial agents. Although the use of antimicrobials in veterinary care has not increased over the last 5 years, it is important to implement and maintain a minimal use strategy from here on. In the human health system in Bhutan, there is an established antimicrobial stewardship program in which each antimicrobial on the Essential Medicines List is classified as Access, Watch, or Reserve (AWaRe) to help optimise their use [18]. In animal health, the World Organization for Animal Health (WOAH, previously OIE) has categorised veterinary antimicrobials as important, highly important, or critically important [19], while the Australian Strategic and Technical Advisory Group (ASTAG) has classified both human and veterinary antimicrobials into low, medium, and high importance [20]. The ASTAG has also identified some classes of antimicrobials that should not be used in livestock. Bhutan may adopt the ASTAG classification system to support its veterinary AMS programme and optimise use of antimicrobials in livestock so that veterinarians are provided guidance in the selection of appropriate antimicrobials when treating infections in animals.

Bhutan is currently reporting the use of antimicrobials to WOAH using Option 1—reporting the overall amount sold for use/used in animals by antimicrobial class, with the possibility to separate by type of use. This is the minimal reporting option and has been adopted because data on use by animal group or species and on the route of administration are not available. An online database, the *Veterinary Information System* (VIS), is currently being built and tested. Full operability of the VIS should enable the animal health sector to capture all data on antimicrobial use in real time and enable reporting to WOAH to move from Option 1 to Option 3, where data on overall amounts of antimicrobial use by class, type of use, animal groups, and route of administration are provided. In addition, the animal health sector, in an effort to minimize the use of antimicrobials, should enhance effective implementation of existing disease prevention and control plans to reduce the prevalence of disease outbreaks. The sector can also strengthen farm biosecurity and optimise husbandry practices to reduce disease incursions onto livestock farms. This can be combined with national vaccination targets to achieve >80% vaccination coverage against all economically important livestock and poultry diseases.

The main limitation of this study was the calculation of rates of antimicrobial consumption based on the total animal biomass and the total quantity of antimicrobials rather than segregation of data on each class of antimicrobials available for each livestock species. In the absence of these data, analysis of the rates of consumption at the species level for the different classes of antimicrobials could not be calculated. In the future, this should be able to be addressed using the VIS, once it has been fully implemented.

## 5. Conclusions

The rates of antimicrobial consumption in livestock in Bhutan are considerably lower than in other countries in the South Asian region. This demonstrates that imposition of appropriate regulation of importation and use of antimicrobials in animal health, coupled with development of local guidelines for appropriate use, have the potential to achieve major reductions in antimicrobial use in animal health across Asia. Data are currently not available to assess the impact that these differences in antimicrobial consumption have had on patterns of antimicrobial resistance in bacteria in livestock in Bhutan. However, this study has provided a basis for assessment of the impact of strong antimicrobial governance in animal health in future studies on AMR in pathogens of importance in humans and livestock.

## Figures and Tables

**Figure 1 antibiotics-12-00411-f001:**
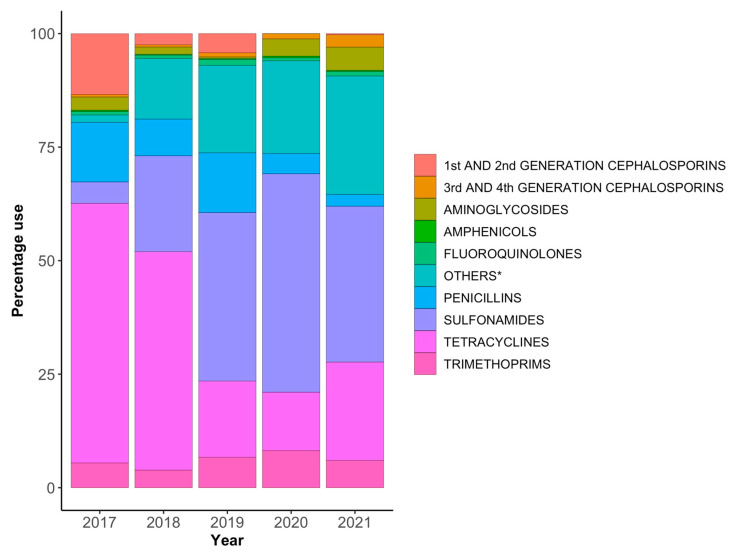
Proportions of different classes of antimicrobials consumed in livestock each year. * Metronidazole and nitrofurazone.

**Table 1 antibiotics-12-00411-t001:** Standardised average weights of different livestock species.

Species	Type	Standard Average Weight (kg)	Description
Bovine *	Adult bovine	425	Average weight of all adult animals
	Heifer bovine	200	Also includes young bulls
	Young bovine	140	Up to the age of 1 year
Horse	Adult	400	Does not include foals
Pig		65	At slaughter age
Sheep		20	At slaughter age
Goat		20	At slaughter age
Poultry	Chickens	1	Includes pullets and adults

* Includes cattle, mithun, yaks, and buffalos for ease of calculation.

**Table 2 antibiotics-12-00411-t002:** Livestock populations in Bhutan by animal species and type.

Species	Type	2017	2018	2019	2020	2021
Bovines	Adult	229,861	241,647	232,032	213,754	226,455
	Heifer	52,754	52,710	51,373	50,539	62,765
	Young	72,021	75,109	71,934	69,087	56,759
Horses	Adult	18,211	17,103	16,792	14,649	12,418
Pigs		18,815	24,342	20,070	17,577	22,954
Sheep		10,444	10,858	11,466	10,793	10,694
Goats		42,689	52,227	47,735	44,119	59,577
Poultry	Chickens	1,118,178	1,144,746	1,299,810	1,383,714	1,384,449
**Total**		**1,562,973**	**1,618,742**	**1,751,212**	**1,804,232**	**1,836,071**

Data source: NSB, Livestock population, 2017–2021.

**Table 3 antibiotics-12-00411-t003:** Calculated biomass (in 1000 tonnes) of domestic livestock in Bhutan.

Species	Type	2017	2018	2019	2020	2021
Bovines	Adult	97.75	102.85	98.60	90.95	96.05
	Heifer	10.6	10.60	10.20	10.20	12.60
	Young	10.08	10.50	10.08	10.08	7.98
Horses		7.20	6.80	6.80	6.00	7.20
Pigs		1.24	1.56	1.30	1.17	1.50
Sheep/goats		1.06	1.26	1.18	1.10	1.40
Chickens		1.12	1.15	1.30	1.38	1.38

**Table 4 antibiotics-12-00411-t004:** Consumption of different classes of antimicrobials by livestock in Bhutan.

Class	Amounts Consumed (kg)				
	2017	2018	2019	2020	2021
Aminoglycosides	14.63	9.82	2.10	17.58	25.03
Amphenicols	1.43	1.31	1.20	1.60	1.20
1st- and 2nd-generation cephalosporins	66.76	15.08	24.02	0.00	1.11
3rd- and 4th-generation cephalosporins	2.81	2.99	5.23	5.42	13.36
Fluoroquinolones	3.74	4.23	7.23	3.09	4.85
Penicillins	65.40	49.65	74.94	20.68	12.63
Sulphonamides	23.56	129.51	211.18	222.52	167.51
Trimethoprim	27.21	23.74	38.12	37.76	29.30
Tetracyclines	285.15	295.10	95.58	59.49	105.58
Others *	8.17	82.01	109.60	94.33	127.31
Total (kg)	498.86	613.43	569.20	462.47	487.89

* Metronidazole and nitrofurazone.

**Table 5 antibiotics-12-00411-t005:** Overall annual antimicrobial consumption by livestock.

	2017	2018	2019	2020	2021
Total mass of antimicrobials (kg)	498.86	295.10	569.20	462.47	487.89
Livestock population size (PCU × 1000)	129.05	134.72	129.46	120.88	128.11
Rate of antimicrobial consumption (including horses)	3.87	4.55	4.40	3.83	3.81
Rate of antimicrobial consumption (excluding horses)	4.09	4.80	4.64	4.03	4.04

**Table 6 antibiotics-12-00411-t006:** Rates of consumption of different classes of antimicrobials in livestock in Bhutan.

Class	Rate of Consumption (mg/PCU)				
	2017	2018	2019	2020	2021
Aminoglycosides	0.11	0.07	0.02	0.15	0.20
Amphenicols	0.01	0.01	0.01	0.01	0.01
1st- and 2nd-generation cephalosporins	0.52	0.11	0.19	0.00	0.01
3rd- and 4th-generation cephalosporins	0.02	0.02	0.04	0.04	0.10
Fluoroquinolones	0.03	0.03	0.06	0.03	0.04
Penicillins	0.51	0.37	0.58	0.17	0.10
Sulphonamides	0.18	0.96	1.63	1.84	1.31
Trimethoprim	0.21	0.18	0.29	0.31	0.23
Tetracyclines	2.21	2.19	0.74	0.49	0.82
Others *	0.06	0.61	0.85	0.78	0.99

* Metronidazole and nitrofurazone.

**Table 7 antibiotics-12-00411-t007:** Annual expenditure on antimicrobials for livestock in Bhutan.

Class	Cost (USD)				
	2017	2018	2019	2020	2021
Aminoglycosides	11,587	3067	4041	4146	2859
Amphenicols	432	455	453	604	453
Cephalosporins	13,429	3531	6060	2319	5484
Fluoroquinolones	1095	1133	3121	1649	1816
Penicillins	11,330	11,457	9261	9766	17,977
Sulphonamides/trimethoprim	15,293	20,719	42,848	36,224	30,399
Tetracyclines	17,753	14,146	8551	1516	9554
Others *	3008	4628	2203	3723	3476
Total	73,927	59,135	76,538	59,948	72,017

* Metronidazole and nitrofurazone.

## Data Availability

All data are publicly available at the sites referred to in the paper or upon request from the authors.

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
