# Peer review of "Antimicrobial Consumption in the Livestock Sector in Bhutan: Volumes, Values, Rates, and Trends for the Period 2017–2021"

_antibiotics, 2023, doi:10.3390/antibiotics12020411_

Round 1

Reviewer 1 Report

The article on using and consumption of veterinary antimicrobials in Bhutan in 2017-2021 is fascinating.

It discloses the excellent management given to antibiotics with a lower PCU than in European countries and 120-fold lower in South Asian countries.

It is a very well-written and interesting article; however, it needs more practical application.

The objective is partially done since areas for improvement to reduce the use of antibiotics in livestock still need to be identified.

Conclusions are missing

A change of approach is needed, where the good decisions made in Bhutan can be applied to other countries.

Recommendations, achievements etc., that can help certain countries reduce the use of antibiotics.

The information from PCU Vs. other countries are contrasted but do not offer any recommendation to avoid the excessive use of antibiotics.

Author Response

We have now included a conclusion that emphasizes the impact that strong governance in animal health can have on antimicrobial consumption in livestock, and suggesting that this needs to be considered across South Asia.

Reviewer 2 Report

This study aimed to assess the consumption of veterinary antimicrobials in Bhutan. Data on the use of antimicrobials in livestock may provide evidence to guide policy changes to mitigate the risk of antimicrobial resistance (AMR)

Author Response

No response required.

Reviewer 3 Report

Dear Authors,

Structurally, the article lacks lateral numbering, which makes it difficult to observe reviews and certain defining elements for Antibiotics journal, and in general for any MDPI publication: keywords, authors' contributions, conflicts of interest and other ethical statements.

The topic of this statistical study is of zonal interest, even if the statistics include areas from Asia or Europe.

In the summary section, a brief reference is made to antibiotic resistance, which may be the consequence of the abuse of antibiotics in animals that reach human consumption.

Page 3, subchapter 2.3: The observation "The consumption of veterinary antimicrobials in Bhutan for 2020 was then compared with that of 31 European nations, other regional countries and New Zealand" requires citation. The consumption of antibiotics in Bhutan as well as the comparisons with European countries are relevant for the study topic. Instead, the study should have finality, the authors should continue this study, in the sense of completion: what is the purpose of the study in fact. It is important to link the consequences of the consumption of antibiotics in animals, for example the prevalence of antibiotic resistance in the area, the occurrence of certain pathologies (for example the sensitivity to certain classes of antibiotics of the population in the area, obesity or other disorders) related to the consumption of treated animals with the most used antibiotics, as it results from the presented statistics.

There should be citation and development on the topic of antibiotic resistance related to the consumption of antibiotics in animals and the persistence of antibiotics used in animal use, in human food. To support this observation, I suggest a few articles that deal with the topic of antibiotic persistence, which ultimately influences the appearance of resistance:

https://doi.org/10.3390/antibiotics11040451.

doi: 10.15386/mpr-1742. Epub 2020 Jul 22.

https://doi.org/10.3390/biomedicines10051121

DOI: 10.1016/j.scitotenv.2018.06.314.

Page 7: "In the human health system in Bhutan, there is an established antimicrobial stewardship program, in which each antimicrobial on the Essential Medicines List is classified as Access, Watch or Reserve (AWaRe) to help optimize their use [14]." For this observation the authors should develop, it may be of international interest this program from Bhutan. 

In my opinion, the authors should insert a subchapter regarding the connection between the consumption of antibiotics in animal feed, very well presented in the article, with the incidence of antibiotic resistance and the prevalence of some pathologies related to this consumption, as I specified in these observations.

Best regards

Author Response

We are sorry we omitted line numbering from the manuscript. We have provided them in the revised version. The keywords and other information you have requested were included in the online submission form.

We believe this study is of broader interest than just the South Asian region. Antimicrobial use in livestock is projected to rapidly rise across Asia over the next 10 years, with many of the global hotspots for use predicted to be in this regions. This projected increase in use has global implications. Demonstration that strong governance can effectively limit use in at least one South Asian country provides an example that could be followed across the region, to global benefit. We have added a conclusion to emphasise this point.

We have now included the citations for the NZ and European data in the Materials and Methods.

While we understand the reviewer’s desire to link the data on antimicrobial use in livestock to rates of resistance or disease, these data are not available for Bhutan at present, and we believe it is better that we avoid speculation on the effects of the lower rates of use in Bhutan on human or animal health. While the references provided by the reviewer on antimicrobial persistence are interesting, we don’t feel they are directly relevant to the conclusions we are able to draw from the data we have presented.

Reviewer 4 Report

Dear Authors,

The study you presented is very interesting and although it concerns only Bhutan, it can be of interest to a wide range of readers, due to the fact that antimicrobial consumption in various sectors and its impact on antimicrobial resistance is a wordlwide problem.

I read carefully your manuscript and I actually have no critical remarks. The study is well designed, the results are clearly presented and the discussion is scientifically sound.

The English language used is very clear and I didn't find any mistakes (or at least those that would be obvious).

You may only consider adding a short Conclusions section summarizing your findings.

Author Response

We have now included a conclusion that emphasizes the impact that strong governance in animal health can have on antimicrobial consumption in livestock.

Round 2

Reviewer 1 Report

“identify areas for improvement to reduce the use of antimicrobials in livestock production systems”------à Which part of the discussion and conclusion is this objective contrasted? If not, please add it.

Author Response

We believe we have identified areas for improvement to reduce the use of antimicrobials in livestock production systems

On lines 270-276 we say "the Australian Strategic and Technical Advisory Group (ASTAG) has classified both human and veterinary antimicrobials into low, medium and high importance [16]. The ASTAG has also identified some classes of antimicrobials that should not be used in livestock. Bhutan may adopt the ASTAG classification system to support its veterinary AMS programme and optimise use of antimicrobials in livestock, so that veterinarians are provided guidance in the selection of appropriate antimicrobials when treating infections in animals.

And

On lines 287-293 we say "In addition, the animal health sector, in an effort to minimize the use of antimicrobials, should enhance effective implementation of existing disease prevention and control plans to reduce the prevalence of disease outbreaks. The sector also can strengthen farm biosecurity and optimise husbandry practices to reduce disease incursions onto livestock farms. This can be combined with national vaccination targets to achieve > 80% vaccination coverage against all economically important livestock and poultry diseases.

Reviewer 3 Report

Dear authors

The relationship between the consumption of antibiotics and their persistence is extremely relevant, but for the reasons you stated, you do not consider it necessary to include this link, which represents your point of view, which I respect.

Best regards

Author Response

Thank you. No further revision is requested.